# Rotenone Blocks the Glucocerebrosidase Enzyme and Induces the Accumulation of Lysosomes and Autophagolysosomes Independently of LRRK2 Kinase in HEK-293 Cells

**DOI:** 10.3390/ijms241310589

**Published:** 2023-06-24

**Authors:** Laura Patricia Perez-Abshana, Miguel Mendivil-Perez, Carlos Velez-Pardo, Marlene Jimenez-Del-Rio

**Affiliations:** Neuroscience Research Group, Institute of Medical Research, Faculty of Medicine, University of Antioquia, University Research Headquarters, Calle 62#52-59, Building 1, Laboratory 411/412, Medellin 050010, Colombia; lpatricia.perez@udea.edu.co (L.P.P.-A.); miguel.mendivil@udea.edu.co (M.M.-P.); calberto.velez@udea.edu.co (C.V.-P.)

**Keywords:** autophagy, conduritol-β-epoxide, glucocerebrosidase, HEK-293, LRRK2, Parkinson’s, rotenone

## Abstract

Parkinson’s disease (PD) is a neurodegenerative disorder caused by the progressive loss of dopaminergic (DAergic) neurons in the substantia nigra and the intraneuronal presence of Lewy bodies (LBs), composed of aggregates of phosphorylated alpha-synuclein at residue Ser^129^ (p-Ser^129^α-Syn). Unfortunately, no curative treatment is available yet. To aggravate matters further, the etiopathogenesis of the disorder is still unresolved. However, the neurotoxin rotenone (ROT) has been implicated in PD. Therefore, it has been widely used to understand the molecular mechanism of neuronal cell death. In the present investigation, we show that ROT induces two convergent pathways in HEK-293 cells. First, ROT generates H_2_O_2_, which, in turn, either oxidizes the stress sensor protein DJ-Cys^106^-SH into DJ-1Cys^106^SO_3_ or induces the phosphorylation of the protein LRRK2 kinase at residue Ser^395^ (p-Ser^395^ LRRK2). Once active, the kinase phosphorylates α-Syn (at Ser^129^), induces the loss of mitochondrial membrane potential (ΔΨ_m_), and triggers the production of cleaved caspase 3 (CC3), resulting in signs of apoptotic cell death. ROT also reduces glucocerebrosidase (GCase) activity concomitant with the accumulation of lysosomes and autophagolysosomes reflected by the increase in LC3-II (microtubule-associated protein 1A/1B-light chain 3-phosphatidylethanolamine conjugate II) markers in HEK-293 cells. Second, the exposure of HEK-293 LRRK2 knockout (KO) cells to ROT displays an almost-normal phenotype. Indeed, KO cells showed neither H_2_O_2_, DJ-1Cys^106^SO_3,_ p-Ser^395^ LRRK2, p-Ser^129^α-Syn, nor CC3 but displayed high ΔΨ_m_, reduced GCase activity, and the accumulation of lysosomes and autophagolysosomes. Similar observations are obtained when HEK-293 LRRK2 wild-type (WT) cells are exposed to the inhibitor GCase conduritol-β-epoxide (CBE). Taken together, these observations imply that the combined development of LRRK2 inhibitors and compounds for recovering GCase activity might be promising therapeutic agents for PD.

## 1. Introduction

Parkinson’s disease (PD) is a neurodegenerative disorder characterized by motor symptoms such as bradykinesia, rigidity, resting tremor, and gait disturbance [1]. PD is mainly caused by the progressive loss of dopaminergic (DAergic) neurons in the substantia nigra and the intraneuronal presence of Lewy bodies (LBs), composed of aggregates of alpha-synuclein (α-Syn) [2]. Although initially described in six subjects [3], PD has reached pandemic proportions [4]. Indeed, it is projected that 12 million people will suffer from the neurologic disorder by 2040 [5], mainly affecting the population under 65 years of age [6]. Unfortunately, no curative treatment is available yet. To further aggravate matters, the etiopathogenesis of the disorder is still unresolved. Despite this drawback, mitochondrial damage, oxidative stress (OS), and alteration in the autophagy–lysosomal pathway (ALP) have been suggested to be important determinants in the development of PD [7,8,9]. Interestingly, neurotoxins targeting mitochondrial complex I, such as MPTP [10,11] and rotenone (ROT) [12], have been implicated in PD. Indeed, ROT is a naturally occurring organic heteropentacyclic compound, chemically known as Benzopyrano(3,4-b) furo(2,3-h) (1) benzopyran-6 (6aH)-one, 1,2,12,12a-tetrahydro-2-a-isopropenyl-8,9-dimethoxy (PubChem CID 6758), and is mainly found in the roots of the *Derris* [13,14] and *Lonchocarpus* [15] plant species. It is used worldwide due to its broad-spectrum insecticidal, acaricidal, and pesticidal properties (http://www.chm.bris.ac.uk/motm/rotenone/ accessed in June 2023). Importantly, ROT induces the specific degeneration of DAergic neurons in vitro and in vivo [16], which intrinsically deteriorates in Parkinson’s disease (PD) [17]. ROT works as a strong inhibitor of complex I of the mitochondrial respiratory chain [18] via the inhibition of electron transfer from the iron–sulfur centers in complex I to ubiquinone, leading to the overproduction of reactive oxygen species (ROS) such as superoxide anion radical (O_2_^−^), which dismutates into hydrogen peroxide (H_2_O_2_) [19]. Interestingly, genetic forms of PD affect different mitochondria-associated proteins, such as leucine-rich repeat serine/threonine protein kinase 2 (LRRK2), as well as autophagy–lysosomal proteins (e.g., α-Syn; glucosylceramidase beta 1 (GBA1)), trigger mitochondrial alterations, OS, autophagy, and lysosomal dysfunction (e.g., [20,21,22]). Nonetheless, how LRRK2, α-Syn, and GBA1 proteins work alone or interact with each other during a loss of DAergic neurons is not yet clear [23]. Understanding their mechanism of action might be of therapeutic importance for PD.

LRRK2 (Online Mendelian Inheritance in Man (OMIM) #609007) is a large multidomain protein (2527 amino acids and MW 286 kD), composed of a leucine-rich repeat (LRR) domain, a kinase domain, among 5 other important domains (Uniprot protein accession #Q5S007). Therefore, it is not a surprise that LRRK2 would be involved in several cellular processes, including mitochondrial dysfunction, autophagy, OS signaling, and cell death [24]. Recently, we have demonstrated that ROT, a widely used neurotoxin to model PD [16,25], induces a significant increase in intracellular reactive oxygen species (ROS) such as H_2_O_2_, triggers the phosphorylation of LRRK2 (at residue Ser^935^) and c-JUN (at residue Ser^63^/Ser^73^), enhances the expression of the proteins TP53 and p53 upregulated modulator of apoptosis (PUMA), produces cleaved caspase 3 (CC3), induces DNA fragmentation, and decreases mitochondrial membrane potential (ΔΨ_m_) in nerve-like cells (NLCs) compared to untreated cells [26]. Importantly, the LRRK2 kinase inhibitor PF-06447475 protects NLCs against ROT-induced toxic effects. The inhibitor not only blocks the p-S^935^ LRRK2 kinase but also completely abolishes ROS and significantly reverses all ROT-induced apoptosis signaling and OS-associated markers to comparable control values. Taken together, these observations support the hypothesis that LRRK2 functions as a pro-apoptotic kinase under OS [26]. However, how LRRK2 would affect α-Syn in our experimental paradigm has not been determined. Some researchers have suggested that LRRK2 might directly or indirectly interact with α-Syn [27]. Of note, it has been shown that α-Syn aggregation is enhanced by LRRK2 in human neuroblastoma SH-SY5Y cells [28]. Therefore, whether LRRK2-induced apoptosis occurs dependently or independently of α-Syn is still an open question.

Several studies have demonstrated that PD-associated genes related to the autophagy–lysosomal pathway, such as LRRK2 and GBA1, among others, are involved in the dysfunction of the cellular clearance system in PD pathogenesis. Indeed, the failure of cellular protein degradation systems plays a major role in α-Syn aggregation [29]. Of importance, defects in lysosomal function result in lysosomal storage disorders (LSDs) involving a neurologic component [30,31]. One of the most common LSDs is Gaucher disease (GD, OMIM #230800) [32]. GD results from a biallelic loss of function of the lysosomal enzyme β-glucocerebrosidase (GCase), encoded by the *GBA1* gene, which is localized in chromosome 1q21 (https://www.genecards.org/; accessed in June 2023). Biochemically, GCase hydrolyzes the substrate glucosylceramide (GlcCer) by cleaving a glucose moiety off the molecule, creating the products glucose (Glc) and ceramide (Cer). Therefore, the intracellular accumulation of GlcCer is responsible for the characteristic “Gaucher cells”, which are of mononuclear phagocyte origin [33]. Interestingly, mutations in *GBA1* are among the most known genetic risk factors for the development of PD [34]. Indeed, there is accumulating evidence that a buildup of GlcCer due to the dysfunction of GCase can also increase the accumulation of α-Syn [35,36]. Furthermore, LRRK2 kinase has been suggested to regulate GCase levels and enzymatic activity differently depending on the cell type in PD [37]. Since the GBA1 pathway might be convergent to LRRK2 and α-Syn, GBA1 has become a potential therapeutic target to slow PD [38]. However, the mechanism by which LRRK2 and α-Syn are associated with dysfunctional GCase, autophagy, and cell death has not yet been fully established.

Autophagy is a complex process that involves the fusion of autophagosomes and lysosomes to form the autophagolysosome to remove superfluous and damaged organelles (e.g., dysfunctional mitochondria) and cytosolic proteins [39]. Autophagy appears as a protective mechanism in response to stress [40,41], and it may or may not be associated with cell death, depending on the intensity of the insult. Such dynamic flux in the formation of the autophagy–lysosome can be modulated by inhibitors such as bafilomycin A1 (BafA1), which inhibits vacuolar-type H^+^-ATPase [42], and chloroquine (CQ), which blocks autophagosome fusion with the lysosome and slows down lysosomal acidification [43,44]. Interestingly, it has been reported that ROT blocks autophagic flux prior to inducing cell death [45]. However, it is not entirely clear if LRRK2 kinase is involved in such altered processes in cells under OS. Moreover, the inhibition of LRRK2 kinase activity results in increased GCase activity in DAergic neurons derived from PD patients with either *LRRK2* or *GBA1* mutations [46]. Yet, it is not yet known whether null *LRRK2* may have a similar effect on HEK-293 cells exposed to ROT.

To acquire an understanding of these issues, the present investigation aimed to investigate the effect of the inhibitor of GCase conduritol-β-epoxide (CBE) and ROT on HEK-293 related to LRRK2, α-Syn, autophagy, and apoptosis. To achieve this aim, HEK-293 LRRK2 WT cells and HEK-293 LRRK2 knockout (KO) cells were used. By means of different techniques of biochemistry, immunofluorescence microscopy, and flow cytometry, we found that CBE and ROT inhibited the enzymatic activity of GCase to a similar extent. Interestingly, ROT and CBE induced a high accumulation of lysosomes and autophagolysosomes in HEK-293 cells, but ROT diminished ΔΨ_m_, induced p-Ser^935^ LRRK2 concomitantly with p-Ser^129^α-Syn, and induced DJ-1Cys^106^SO_3_ and CC3 in those cells. On the other hand, ROT inhibited GCase in HEK-293 LRRK2 KO cells. Consequently, it induced a high accumulation of lysosomes and autophagolysosomes but was ineffective in triggering damage to ΔΨ_m_, p-Ser^395^ LRRK2p-Ser^129^α-Syn, DJ-1Cys^106^SO_3_, and CC3. Taken together, these results suggest that ROT decreases the activity of GCase, induces mitochondrial damage, phosphorylates LRRK2, which, in turn, phosphorylates α-Syn, triggers the concomitant accumulation of lysosomes and autophagolysosomes, and causes signs of OS and apoptosis in HEK-293 cells. Of note, ROT impairment of ALP and the induction of apoptosis occur in an LRRK2-independent and LRRK2-dependent fashion, respectively, supporting the idea that LRRK2 is a pro-apoptotic kinase in cells under OS stimuli. Therefore, LRRK2 has become a therapeutic target for the treatment of PD.

## 2. Results

### 2.1. Rotenone (ROT) Inhibits Glucocerebrosidase (GCase) Activity by Mimicking the Inhibitor Conduritol-β-Epoxide (CBE) in HEK-293 Cells

CBE is a cyclitol epoxide that covalently and irreversibly reacts with the catalytic nucleophile of the lysosomal enzyme GCase and, thus, irreversibly inactivates the enzyme. We, therefore, first evaluated whether CBE inhibits GCase in HEK-293 cells. Effectively, Figure 1 shows that the enzymatic activity of GCase decreased by −62% and −87% in HEK-293 exposed to (10 μM) and (50 μM) CBE, respectively, compared to untreated cells (Figure 1A). An in silico molecular docking simulation analysis [47] revealed that CBE binds to a pocket in GCase (Protein Data Bank, PDB #6T13, Vina score: −6.0) interacting with at least 15 residues (Table 1), wherein the residue Glu^340^ of the protein GCase is the catalytic nucleophile critical for covalent binding with the inhibitor (Figure 1B and inset) [48]. Interestingly, HEK-293 cells exposed to (10 μM) ROT or (50 μM) ROT for 6 h diminished the catalytic activity of GCase by −48% and −67%, respectively, compared to untreated cells (Figure 1C). Like CBE, molecular docking analysis predicted that ROT would bind to similar amino acid residue pockets in GCase with 94% (14/15) amino acid similarity, with a Vina score of −9.2 (Table 1), including residue Glu^340^ (Figure 1D and inset). Given that (10 μM) ROT significantly reduced GCase, we selected this concentration for further experiments.

### 2.2. Rotenone (ROT) and Conduritol-β-Epoxide (CBE) Induce Accumulation of Lysosomes but ROT Affects the Mitochondrial Membrane Potential (ΔΨm) Only in HEK-293 Cells

Given that GCase is localized in lysosomes [49], we wanted to evaluate the effect of ROT and CBE on lysosomes and mitochondrial functionality in HEK-293 cells. To achieve this aim, cells were exposed to CBE (10 and 50 μM) or ROT (10 μM) for 24 h. Since LysoTracker Deep Green is a cell-permeable, non-fixable, green, fluorescent dye that stains acidic compartments within a cell, we were able to unveil the cellular granular content of the cell, i.e., lysosomes, which are acidic cellular compartments in nature. Flow cytometry analysis shows that CBE (10 and 50 μM) and ROT (10 μM) induced the accumulation of lysosomes by +77% (Figure 2B), +100% (Figure 2C), and +127% (Figure 2D), respectively, compared to untreated HEK-293 cells (Figure 2A,E). However, while CBE did not perturb ΔΨ_m_ at any concentration tested (Figure 2F–H), ROT reduced ΔΨ_m_ by −20% in HEK-293 cells (Figure 2I,J). Similar data were obtained by fluorescence microscopy (MF) analysis (Figure 2K–P). Based on these observations, and because there was no statistically significant difference between 10 and 50 μM CBE (Figure 2A), we selected 10 μM CBE for further experiments.

### 2.3. Rotenone (ROT) and Conduritol-β-Epoxide (CBE) Induce Both the Accumulation of Lysosomes and an Increase in Autophagolysosomes in HEK-293 Cells

Next, we investigated whether ROT and CBE affect the autophagolysosomal flux in HEK-293 cells. Figure 3 shows that both CBE (10 μM) and ROT (10 μM) induced not only an increase in the accumulation of lysosomes by +80% (Figure 3B) and +127% (Figure 3C), respectively, when compared to untreated cells (Figure 3A,F), but also a significant increase in autophagy–lysosome vacuoles by +138% (Figure 3H) and +252% (Figure 3I), respectively, compared to the control (Figure 3G,L). As mediators of the autophagy flux, HEK-263 cells were exposed to the classic inhibitors of autophagy, chloroquine (CQ) and bafilomycin A1 (BAF), which block the binding of autophagosomes to lysosomes by altering the acidic environment of lysosomes [50]. As expected, CQ (Figure 3D) and BAF (Figure 3E) increased the accumulation of lysosomes by +140% and +67%, respectively, compared to untreated cells (Figure 3A,F), but significantly reduced the formation of autophagolysosomes by −14% (Figure 3J,L) and −52% (Figure 3K,L), respectively. Additionally, we also detected the accumulation of the autophagy marker microtubule-associated protein-light chain 3-II (LC3-II), which is a cytosolic form of LC3 (LC3-I) conjugated to phosphatidylethanolamine to form LC3-phosphatidylethanolamine conjugate (LC3-II) and recruited to autophagosomal membranes [51]. As shown in Figure 3, CBE (Figure 3N), ROT (Figure 3O), CQ (Figure 3P), and BAF (Figure 3Q) induced a statistically significant increase in LC3-II compared to untreated cells (Figure 3M,R). Of note, ROT-induced LC3-II accumulation was equivalent to that induced by CQ, whereas CBE-induced LC3-II accumulation was comparable to BAF (Figure 3R).

### 2.4. Rotenone (ROT) but Not Conduritol-β-Epoxide (CBE) Induces the Oxidation of Stress Sensor Protein DJ-1 and Cleaved Caspase 3 (CC3) in HEK-293 Cells

It is known that the oxidation of the stress sensor protein DJ-1Cys^106^-SH (sulfhydryl group) into DJ-1Cys^106^-SO_3_ (sulfonic acid) is a specific target of the non-radical ROS H_2_O_2_ [52]. We, thus, determined whether CBE or ROT can generate H_2_O_2_ and induce the generation of CC3. Therefore, HEK-293 cells were exposed to CBE (10 μM) or ROT (10 μM) for 24 h. As shown in Figure 4A, while CBE oxidized DJ-1 at a similar percentage as untreated cells, ROT induced a statistically significant increase in oxidized DJ-1 (+440%) compared to untreated cells. Similar observations were found by IMF analysis (Figure 4B–E, 48–f.i. oxDJ-1). On the other hand, ROT induced a statistically significant increase in CC3 (+500%), whereas the percentage of CC3 induced by CBE remained at basal levels comparable to untreated HEK-293 cells (Figure 4F). These observations were confirmed by IMF analysis (Figure 4G–J, 83–f.i. CC3+).

### 2.5. Rotenone (ROT) but Not Conduritol-β-Epoxide (CBE) Induces Phosphorylation of Alpha-synuclein (α-Syn) and LRRK2 Kinase in HEK-293 Cells

We wanted to assess whether ROT and CBE trigger the phosphorylation of α-Syn concurrently with LRKK2 in HEK-293 cells. As shown in Figure 5, ROT but not CBE induced a statistically significant increase in p-Ser^129^α-Syn, as detected by flow cytometry (Figure 5A, +580%), and IMF (Figure 5B–E, 44.5-f.i.). Previously, it was shown that ROT induced the phosphorylation of LRRK2 in nerve-like cells [26]. Therefore, we evaluated whether ROT and CBE cause p-Ser^935^ LRKK2 in HEK-293 cells. Figure 5F shows that ROT induced p-Ser^935^ LRKK2 by +1580% in HEK-293 cells, as evaluated by flow cytometry. In contrast, CBE was unable to induce any effect on LRRK2 (Figure 5F). Similar data were documented by IMF (Figure 5G–J, 4.4–f.i. ROT > p-S^935^ LRRK2).

### 2.6. Rotenone (ROT) Does Not Induce the Phosphorylation of LRRK Kinase in HEK-293 LRRK2 Knockout (KO) Cells

The above finding that ROT induced the phosphorylation of LRRK2 prompted us to expand our observation by inquiring whether ROT could induce p-Ser^395^ LRRK2 in HEK-293 LRRK2 KO cells. We, therefore, first confirmed the protein expression status of LRRK2 in both wild-type (WT) HEK-293 and HEK-293 LRRK2 KO cells. Figure 6 shows the expression of LRRK2 in WT HEK-293 cells, but its expression was almost completely reduced in HEK-293 LRRK2 KO cells, according to flow cytometry analysis (−90%, Figure 6A). When both WT and KO cells were exposed to ROT, it was observed that the neurotoxin induced p-Ser^395^ LRRK2 by +820% in WT HEK-293 cells (Figure 6B,C), but this effect was drastically reduced by −96% in HEK-293 LRRK2 KO cells when compared to treated WT cells (Figure 6B,C). Similar observations were obtained by IMF analysis (Figure 6E–H).

### 2.7. ROT Inhibits the Enzymatic Activity of GCase Equally in Both WT and HEK-293 KO Cells

In parallel experiments, we wanted to determine whether ROT affects the enzymatic activity of GCase in KO cells. To achieve this end, WT and HEK-293 LRRK2 KO cells were first exposed to ROT for 24 h and then quantified for the percentage of enzymatic activity. As shown in Figure 6I, ROT inhibited the activity of GCase to a similar extent in WT and HEK-293 LRRK2 KO cells.

### 2.8. ROT Induces the Accumulation of Lysosomes and Reduces Mitochondrial Potential in HEK-293 KO Cells

Then, we wondered whether ROT could alter the lysosomal system and damage ΔΨm in KO cells. Flow cytometry analysis revealed that untreated WT and KO cells showed no statistical difference in the percentages of lysosomal accumulation (Figure 7A,C,E) or the loss of ΔΨm (Figure 7F,G). In contrast, ROT induced not only a significant accumulation of lysosomes in both WT HEK-293 cells (Figure 7B,E) and KO cells (Figure 7D,E), but induced a significant decrease in mitochondrial functionality reflected by low ΔΨ_m_ in WT only (Figure 7F,G).

### 2.9. Rotenone (ROT) Induces An Increase in Autophagosomes Independently of LRRK2

To determine whether ROT affects the autophagosome flux in HEK-293 LRRK2 KO cells, WT and KO cells were exposed to ROT for 24 h. As shown in Figure 8, ROT provoked the accumulation of lysosomes (Figure 8B) and increased autophagolysosomes (Figure 8G) in both HEK-293 LRRK2 WT and KO cells (Figure 8E,J) compared to untreated cells (Figure 8A,E,F,J). Similarly, ROT induced the accumulation of LC3-II in both WT and KO cells (Figure 8L,P,S) compared to untreated cells (Figure 8K,O,S). As expected, CQ and BAF induced the accumulation of lysosomes (Figure 8C–E), a significant reduction in autophagosome and lysosome formation (Figure 8H–J), and induced the accumulation of LC3-II (Figure 8M,N,Q–S), irrespective of LRRK2 gene status in HEK-293 cells compared to untreated cells (Figure 8E,J,S).

### 2.10. ROT Neither Induces the Phosphorylation of α-Syn, the Oxidation of DJ-1, Nor the Activation of Caspase 3 (CC3) in HEK-293 LRRK2 KO Cells

We wanted to determine the effect of ROT in KO cells regarding α-Syn, DJ-1, and CC3. As shown in Figure 9, ROT was highly efficient, inducing p-Ser^129^α-Syn by +1000% (Figure 9A,B), oxidizing DJ-1 by +2500% (Figure 9H,I), and the production of CC3 by +1800% (Figure 9O,P) in WT HEK-293 cells. In sharp contrast, ROT was strongly ineffective in triggering the phosphorylation of α-Syn (Figure 9A,B), oxDJ-1 (Figure 9H,I), and CC3 (Figure 9O,P) in HEK-293 LRRK2 KO cells. Similar observations were found by IMF analysis (Figure 9C–G,J–N,Q–U).

## 3. Discussion

We confirm that CBE significantly reduced the enzymatic activity of GCase (e.g., by −87% at 50 μM) via binding to Glu^340^ of the enzyme, as verified by in silico molecular docking analysis [48,53]. Therefore, a deficiency in GCase activity might lead to diminished performance of the enzyme in the lysosome and the disruption of lysosomal targeting. Interestingly, CBE induces the accumulation of lysosomes to a similar extent as the lysosome inhibitors, CQ and BAF, in HEK-293 cells. These observations suggest that CBE is an effective compound to dysregulate the autophagy pathway [54]. Under the present experimental conditions, we found that CBE causes neither mitochondrial potential alteration, triggers OS, as reflected by the non-oxidized sensor protein DJ-1Cys^106^-SOH, impairs ΔΨ_m_, induces α-Syn and LRRK2 phosphorylation, nor produces CC3 in HEK-293 cells. However, we do not discard the possibility that CBE under prolonged incubation might reverse the fate of these cellular proteins and mitochondria [55]. Therefore, we were able to separate malfunctioning autophagy processes (i.e., lysosome from autophagosome fusion) from OS and apoptosis. Surprisingly, ROT diminishes the activity of GCase (by −48% at 10 μM ROT), according to the enzymatic GCase test, most probably through its binding to the critical catalytic residue Glu^340^, as predicted by molecular docking analysis. Accordingly, ROT interacts with 14 out of 15 residues similar (94% similarity) to the ones reported for CBE in the catalytic pocket of GCase. Although ROT displays a much more negative Vina score, which is indicative of a strong binding affinity (e.g., −9.2 ROT versus −6.0 CBE), CBE is much more specific towards GCase than ROT, according to the biochemical reduction in GCase activity. Despite this drawback, ROT proves to be highly effective, provoking, in a simultaneous fashion, a significant increase in the oxidation of DJ-1Cys-SH into DJ-1Cys^106^SO_3_, CC3, the phosphorylation of α-Syn and LRKK2 kinase, the accumulation of both lysosomes and autophagolysosomes, and a significant decrease in ΔΨ_m_. These data imply that ROT triggers (i) the accumulation of lysosomes and autophagolysosomes, (ii) mitochondrial-dependent OS damage, and (iii) apoptosis in HEK-293 cells.

However, how does ROT link these three processes? Our findings suggest that ROT triggers two alternative and complementary mechanisms, involving the interaction between ROT and GCase and ROT and mitochondrial complex I, which eventually converge on apoptosis. Mechanistically, ROT functions as a strong inhibitor of complex I of the mitochondrial respiratory chain [18] via the inhibition of electron transfer from the iron–sulfur centers in complex I to ubiquinone, leading to a blockade of the I_Q_ site [56], and an over-reduction of complex I causes electrons to leak and produce ROS superoxide anion radical (O_2_^−^). The O_2_^−^ radical can dismutase into non-radical reactive H_2_O_2_ [19], which, in turn, via signaling mechanisms [57,58], oxidizes the stress sensor protein DJ-1, leading to the overproduction of DJ-1Cys^106^SO_3_ [52]. Interestingly, oxidized DJ-1 has been proposed as a possible biomarker of PD [59,60]. Alternatively, H_2_O_2_ might activate LRRK2 kinase activity by directly enhancing its autophosphorylation, e.g., at Tyr^1967^ [61], Ser^2032^, and Tyr^2035^ [62,63], or indirectly, via the phosphorylation of Ser^910^ and Ser^935^ by the inhibition of the nuclear factor-κB (IκB) kinase (IKK) complex [64]. Of note, we found p-Ser^935^ LRRK2-positive cells in HEK-293 cells exposed to ROT. Interestingly, it has been found that H_2_O_2_ increases LRRK2 kinase activity and enhances LRRK2 cell toxicity in HEK-293T cultured cells, mouse primary cortical neuronal cultures [65], and nerve-like cells [26]. Therefore, phosphorylated LRRK2 kinase might directly or indirectly damage mitochondria and trigger the activation of several cell-death-related proteins: (i) LRRK2 directly interacts with dynamin-like protein 1 (DLP1), a key mitochondrial fission protein, increasing its mitochondrial targeting and, thus, promoting mitochondrial fragmentation [66]; (ii) increases the phosphorylation of peroxiredoxin 3, exacerbating OS-induced cell death [67]; (iii) phosphorylates both the activation of apoptosis-signal-regulating kinase 1 (ASK1) [68] and MKK4/MAPK kinase [69], thereby triggering the JNK/c-JUN/PUMA death pathway; and (iv) phosphorylates the pro-apoptotic transcription factor TP53 [70], thus triggering downstream apoptosis signaling. Taken together, these observations suggest that phosphorylated LRRK2 might work as a pro-apoptotic kinase under OS stimuli [26,71]. Furthermore, (v) LRRK2 kinase phosphorylates α-Syn at Ser^129^ [72], which is the major component of pathological deposits in PD [73]. In line with this, we found p-Ser^935^ LRRK2 concomitant with p-Ser^129^ α-Syn-positive cells in HEK-293 cells exposed to ROT. Although the exact mechanism by which α-Syn causes dopamine neuronal loss is unclear, α-Syn has been suggested to interfere with mitochondrial dynamics and promote mitochondrial fragmentation through α-Syn- and DLP1-dependent mechanisms or by the direct binding of α-Syn to mitochondria, occurring independently of proteins involved in mitochondrial dynamics [66,74]. All these effects lead to an acceleration in the disposal of damaged mitochondria through mitophagy [75] and/or the activation of the pro-apoptosis protein caspase 3 (this work). Additionally, aggregated α-Syn might disrupt phagosome and lysosome fusion [76]. Indeed, α-Syn has been shown to disrupt intracellular trafficking and the lysosomal activity of GCase [77]. Taken together, these observations suggest a self-propagating positive feedback process in which elevated levels of toxic α-Syn lead to a depletion of lysosomal GCase, resulting in a progressive accumulation of GlcCer that promotes the additional formation of toxic α-Syn [78]. However, we found that there is no p-Ser^129^α-Syn in CBE-exposed cells. In agreement with others [79], this observation suggests that the chemical inhibition of GCase activity per se is not sufficient to provoke the phosphorylation and accumulation of α-Syn, and that the inhibition of GCase has to be accompanied by an additional stimulus, e.g., mitochondrial-derived OS, necessary to trigger the phosphorylation of α-Syn via LRRK2. Indeed, α-Syn can directly inhibit lysosomal GCase activity [80,81], or it can indirectly reduce GCase activity by inhibiting its transport from the endoplasmic reticulum to the lysosomes [82]. As expected, however, CBE only increased intracellular granular content and acidic particles, i.e., lysosomes, and augmented autophagy–lysosome fusion.

We hypothesized that HEK-293 cells carrying an LRRK2 null gene would revert to normal the cytotoxic effects associated with ROT-induced mitochondrial damage, lysosomal dysfunction, impaired autolysosome formation, OS marker DJ-1, and apoptosis marker CC3 in HEK-293 LRRK2 KO cells. Effectively, we found that HEK-293 LRRK2 KO cells exposed to ROT exhibit an accumulation of lysosomes and autophagolysosomes only. Several observations support these findings. First, ROT reduced the levels of GCase activity to similar percentage values in WT and KO cells. Second, no significant changes in ΔΨ_m_ were observed in LRRK2 WT and KO cells. Third, ROT induced almost neither p-Ser^129^α-Syn, oxD-1Cys^106^SO_3_, nor CC3-positive cells in KO cells. Finally, ROT induced a significant increase in the accumulation of lysosomes and autophagolysosomes in WT and LRRK2 KO cells, as reflected by the accrual of the LC3-II marker. Taken together, these results suggest that HEK-293 LRRK2 null cells become resilient to ROT-induced OS and apoptosis, but the mutant cells still suffer from lysosomal and autophagy dysfunctionality, which, under prolonged incubation, may lead to cell death. Interestingly, the pathologic phenotype of HEK-293 LRRK2 KO cells exposed to ROT resembles the phenotype of WT HEK-293 cells exposed to CBE only. Our findings suggest that the dysfunction of lysosomal activity and/or the disturbance of autophagy–lysosome fusion is independent of LRRK2 kinase activity. Thus, these results support the notion that LRRK2 is a critical kinase in the apoptosis pathway and α-Syn is a major mediator of LRRK2-induced toxicity. In contrast to others [46,83], our findings suggest that LRRK2 kinase might not represent a direct regulator of lysosomal GCase, lysosomal function, or autophagy–lysosome fusion. However, we do not discard the possibility that LRRK2 kinase activity affects both the levels and catalytic activity of GCase in a cell-type-specific manner [37]. Further investigation is, therefore, needed to clarify this issue.

HEK-293 cells have been extensively used as an in vitro model system to study PD due not only to their easy handling, reliable growth, and propensity for transfection but also to their amenability to stringent quantitative assessments. Most notably, HEK-293 cells express the typical features of immature neurons, such as the neurofilament (NF) subunits NF-L, NF-M, and NF-H, α-internexin, vimentin, and keratins 8 and 18, and also reveal the expression of mRNAs specific for numerous other genes normally expressed in neuronal lineage cells [84]. Therefore, HEK-293 might qualify as a human neuronal cell line model. Indeed, HEK293 cells provide a reasonable approximation for addressing numerous questions of basic biology in PD. Indeed, HEK-293 cells have demonstrated a clear pro-apoptotic transcriptional response profile similar to that in neurons undergoing apoptosis [85]. Moreover, HEK-293 cells display elements of the autophagy–lysosome that are mechanistically similar to those expressed in DAergic neurons [50]. Interestingly, since LRRK2 is a well-conserved evolutionary gene [86], HEK-293 cells have been used to identify molecular substrates of this kinase [87] and to study LRRK2 mutations’ functional analysis [88,89]. HEK-293 cells have also been used to demonstrate that, depending on its concentration, the neurotoxin ROT can induce sublethal and/or lethal effects. For instance, it has been shown that ROT (10 nM) might induce the cytosolic production of H_2_O_2_ only in HEK-293 cells [90], whereas, at higher concentrations (e.g., 10 μM), it induces autophagy and apoptosis ([45,91] and this work). Last but not least, HEK-293 cells have been used to ectopically express not only the human dopamine transporter (hDAT) to study, e.g., the toxic effect of MPTP [92], but also dopaminergic receptors, e.g., D1 [93] and/or D5 [94]. Taken together, all these biological features suggest that HEK-293 cells are a highly promising cellular model to reveal the molecular aspects, as described in the present investigation, of the insidious degenerative disorder PD.

## 4. Materials and Methods

### 4.1. HEK-293 Cell Line

The HEK-293, a specific immortalized cell line derived from a human embryonic kidney, was purchased from AcceGen Biotech (cat #ABC-TC0008, AcceGen Biotech, Fairfield, NJ, USA), and the HEK-293 LRRK2 knockout (OK) cell line was kindly provided by Dr. F. Martin (Genomic Medicine Department, GENYO, Centre for Genomics and Oncological Research, Pfizer-University of Granada–Andalusian Regional Government, Granada, Spain). Both HEK-293 and HEK-293 LRRK2 KO cells were cultured according to the suppliers’ recommendations. Briefly, cells were grown in Dulbecco’s Modified Eagle’s Medium (DMEM, cat #D0819, Sigma, Saint Louis, MO, USA), supplemented with fetal bovine serum (FBS, cat #CVFSVF00-01, Eurobio Scientific, Paris, France) to a final concentration of 10% in a humidified incubator at 37 °C, supplemented with 5% CO_2_. Growth media were replaced every 2–3 days.

### 4.2. Analysis of Cells

#### 4.2.1. Assay Protocol

HEK-293 cells were cultured in DMEM with low glucose plus 10% fetal bovine serum (FBS) and left untreated or treated with the GCase inhibitor (10, 50 μM) CBE (cat #6090-95-5, Santa Cruz Biotechnology, Dallas, TX, USA) or ROT (10, 50 μM; cat #150154, ICN Biomed, Paris, France) for 6 h. For in vitro inhibition of autophagy, cultured cells were treated with bafilomycin A1 (10 nM, cat no. B1793, Sigma-Aldrich, Saint Louis, MO, USA) or chloroquine (10 μM, cat no. C6628, Sigma-Aldrich, Saint Louis, MO, USA) for 24 h prior to ROT exposure.

#### 4.2.2. GCase Activity Assay

Cellular GCase activity was determined using the Beta-Glucosidase Assay Kit (cat #ab272521, Abcam, Boston, MA, USA) according to the manufacturer’s recommendations. Briefly, cell lysates were incubated with p-nitrophenyl-α-D-glucopyranoside, which is hydrolyzed specifically by β-glucosidase into a yellow-colored product (maximal absorbance at 405 nm). The rate of the reaction was directly proportional to the enzyme activity.

#### 4.2.3. Characterization of Lysosomal Complexity

To analyze lysosomal complexity, cells were incubated with the cell-permeable, non-fixable, green, fluorescent dye LysoTracker Green DND-26 (50 nM, cat #L7526, Thermo Fisher Scientific, Waltham, MA, USA) for 30 min at 37 °C. Cells were then washed, and LysoTracker fluorescence was determined by analysis of fluorescence microscopy images in a Floid Cells Imaging Station microscope (Cat# 4471136, Life Technologies, Carlsbad, CA, USA), or flow cytometry using a BD LSRFortessa II flow cytometer (BD Biosciences, Franklin Lakes, NJ, USA). The experiment was conducted 3 times, and 10,000 events were acquired for analysis. Flow cytometry analysis for LysoTracker/SSCA was performed by selecting, in the FL-1 channel, all cells with LysoTracker reactivity (>99%), in order to perform the analysis of the total LysoTracker-positive population. SSCA parameter was adjusted to the mean fluorescence of the control (UNT; 40 K ± 3.5 K) plus two standard deviations (i.e., values above 47 K). Quantitative data and figures were obtained using FlowJo 7.6.2 Data Analysis Software (BD Biosciences, Franklin Lakes, NJ, USA).

#### 4.2.4. Analysis of Mitochondrial Membrane Potential (ΔΨm)

The assessment of ΔΨm was performed according to Ref. [95]. Briefly, cells were incubated for 20 min at RT in the dark, with a deep-red MitoTracker (20 nM final concentration) compound (cat #M22426, Thermo Scientific, Waltham, MA, USA). Cells were analyzed using fluorescence microscopy Floid Cells Imaging Station microscope (cat# 4471136, Life Technologies, Carlsbad, CA, USA), or a BD LSRFortessa II flow cytometer (BD Biosciences, Franklin Lakes, NJ, USA). The experiment was conducted 3 times, and 10,000 events were acquired for analysis. MitoTracker highly positive cells were selected located between 10^4^ and 10^6^. No discrimination by complexity was made. Quantitative data and figures were obtained using FlowJo 7.6.2 Data Analysis Software (BD Biosciences, Franklin Lakes, NJ, USA).

#### 4.2.5. Detection of oxDJ-1, Cleaved Caspase 3 (CC3), LRRK2, Alpha-Synuclein, and LC3-II Using Fluorescent Microscopy and Flow Cytometry

After each treatment, cells (1 × 10^5^) were fixed in 80% ethanol and stored at 20 °C overnight. Then, cells were washed with PBS and permeabilized with 0.2% triton X-100 (Cat# 93443, Sigma-Aldrich, St. Louis, MO, USA) plus 1.5% bovine serum albumin (BSA, Cat# A9418, Sigma-Aldrich, St. Louis, MO, USA) in phosphate-buffered solution (PBS) for 30 min. Then, cells were washed and incubated with primary antibodies (1:200; diluted in PBS containing 0.1% BSA) against oxidized DJ-1 (1:500; ox (Cys^106^) DJ1; spanning residue C^106^ of human PARK7/DJ1; oxidized to produce cysteine sulfonic (SO_3_) acid; Abcam cat #AB169520; Boston, MA, USA), CC3 (1:250; cat #AB3623, Millipore, Merck, Darmstadt, Germany), p-(S^935^)-LRRK2 (Abcam cat #AB133450; Boston, MA, USA), α-synuclein (pS^129^; Abcam cat #AB51253; Boston, MA, USA), and LC3-II (cat #NB100-2220, Novus Biologicals, Englewood, CO, USA) overnight at 4 °C. After exhaustive rinsing, we incubated the cells with secondary fluorescent antibodies (DyLight 488 horse anti-rabbit and mouse antibodies, cats DI 1094 and DI 2488, Vector Laboratories, Newark, NJ, USA) at 1:500. Finally, cells were washed and re-suspended in PBS for analysis on a BD LSRFortessa II flow cytometer (BD Biosciences, Franklin Lakes, NJ, USA). Twenty thousand events were acquired, and the acquisition analysis was performed using FlowJo 7.6.2 Data Analysis Software (BD Biosciences, Franklin Lakes, NJ, USA). For microscopy, the nuclei were stained with (0.5 μM) Hoechst 33342 (Life Technologies, Carlsbad, CA, USA), and images were acquired on a Floyd Cells Imaging Station microscope (Cat# 4471136, Life Technologies, Carlsbad, CA, USA).

#### 4.2.6. Autophagy Assay

The autophagy assay was performed according to the manufacturer’s recommendation (cat #MAK138, Sigma-Aldrich, Saint Louis, MO, USA). Briefly, cells under different treatments were incubated with 1X stain reagent for 20 min. Then, the fluorescence intensity (ex 360/em 520 nm) was measured using a BD LSRFortessa II flow cytometer (BD Biosciences). Twenty thousand events were acquired, and the acquisition analysis was performed using FlowJo 7.6.2 Data Analysis Software (BD Biosciences, Franklin Lakes, NJ, USA).

### 4.3. Molecular Docking

For in silico molecular docking analysis, we used the X-ray diffraction crystallography protein structure of glucocerebrosidase (GCase; Protein Data Bank (PDB) code: 6T13). Blind molecular docking was performed with CB-Dock version 2 [47], a cavity detection-guided protein–ligand blind docking web server that uses Autodock Vina (version 1.1.2, Scripps Research Institute, La Jolla, CA, USA). The SDF structure files of the tested compounds (conduritol-β-epoxide (CBE): PubChem CID 119054; rotenone (ROT): PubChem CID 6758) were downloaded from PubChem. Molecular blind docking was performed by uploading the 3D structure PDB file of GCase into the server with the SDF file of each compound. For analysis, we selected the docking poses with the strongest Vina score in the catalytical pocket. The generated PDB files of the molecular docking of each compound were visualized with the CB-Dock2 interphase.

### 4.4. Data Analysis

In this experimental design, a vial of HEK-293 (WT and LRRK2 KO cells) was thawed and cultured, and the cell suspension was pipetted at a standardized cellular density of 2 × 10^4^ cells per cm^2^ into different wells of a 24-well plate. Cells (i.e., the biological and observational units) [96] were randomized to wells by simple randomization (sampling without replacement method), and then, wells (i.e., the experimental units) were randomized to treatments using a similar method. Experiments were conducted in triplicate. The data from individual replicate wells were averaged to yield a value of n = 1 for the experiment, and this was repeated on three occasions blind to the experimenter and/or flow cytometer analyst for a final value of n = 3 [96]. Based on the assumptions that the experimental unit (i.e., the well) data comply with independence of observations, the dependent variable is normally distributed in each treatment group (Shapiro–Wilk test), and there is homogeneity of variances (Levene’s test); the statistical significance was determined by one-way analysis of variance (ANOVA) followed by Tukey’s post hoc comparison calculated with GraphPad Prism 5.0 software (https://www.graphpad.com/; accessed on 5 February 2023). Differences between groups were only deemed significant when a *p*-value of 0.05 (*), 0.001 (**), or 0.001 (***). All data were illustrated as the mean SD.

## 5. Conclusions

In this work, we demonstrated that mitochondrial damage, α-Syn, reduced GCase activity, and LRRK2 converge to contribute synergistically to the dysfunction of ALP and apoptosis cell death. Taken together, our findings suggest that ROT induces two convergent pathways. On the one hand, ROT indirectly generates H_2_O_2_ via the inhibition of mitochondrial complex I. H_2_O_2_ not only oxidizes DJ-1Cys^106^SH into DJ-1Cys^106^SO_3_ but also induces p-Ser^935^ LRRK2, which, in turn, phosphorylates α-Syn (p-Ser^129^α-Syn). Interestingly, ROT decreases GCase activity, leading to an accumulation of lysosomes and autophagolysosomes, as evidenced by a significant increase in the accumulation of the LC3-II marker. In parallel, ROT induces the loss of ΔΨ_m_, leading to the production of CC3 and apoptosis-induced cell death (Figure 10A). On the other hand, ROT causes no damage in HEK-293 LRRK2 KO cells. In fact, ROT decreases GCase activity. Consequently, it leads to the accumulation of lysosomes and autophagolysosomes, but there are no signs of OS, e.g., DJ-1Cys^106^SO_3,_ p-Ser^129^α-Syn, damage to ΔΨ_m_, or CC3-positive cells, resulting in cell survival (Figure 10B). Therefore, our findings suggest that LRRK2 might operate as a multi-target pro-apoptotic kinase [26,71], contributing to mitochondrial and cell death concurrently with phosphorylated Ser^129^ α-Syn in cells exposed to OS stimuli [97]. Interestingly, CBE induces, in HEK-293 cells (Figure 10A), a similar cellular phenotype as HEK-293 LRRK2 KO cells exposed to ROT (Figure 10B). These observations imply that the combined development of LRRK2 inhibitors [98,99] and compounds for recovering GCase activity [38,100] might be promising therapeutic agents for PD.

## Figures and Tables

**Figure 1 ijms-24-10589-f001:**
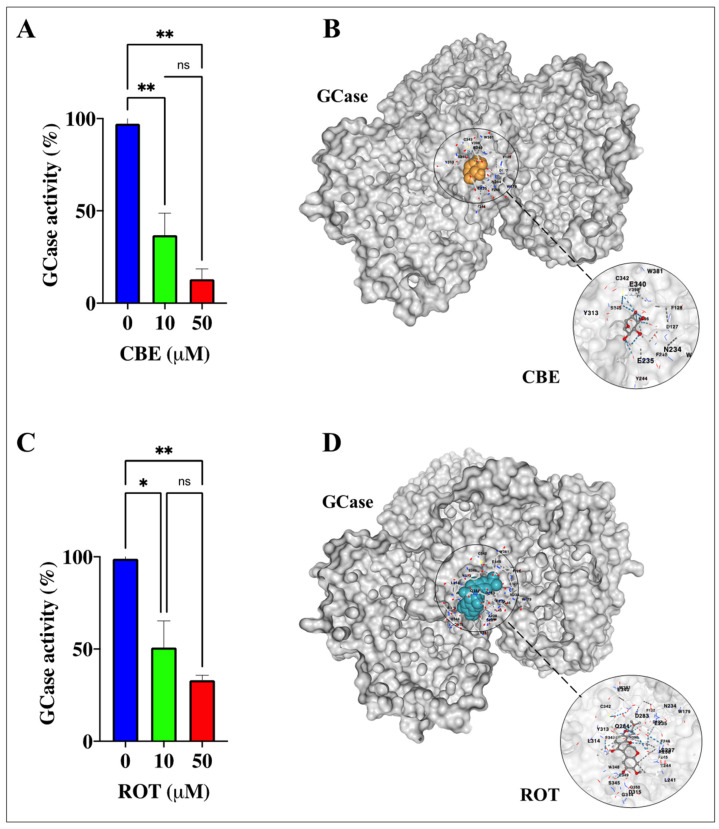
Evaluation of glucocerebrosidase (GCase) activity in HEK-293 cells exposed to conduritol-β-epoxide (CBE) and rotenone (ROT): a molecular docking analysis. (**A**) Analysis of GCase activity in HEK-293 cells exposed to CBE (0, 10, and 50 μM). (**B**) Representative CB-Dock2 3D images showing the molecular docking of GCase (PDB: 6T13) with CBE (PubChem CID 119054). (**C**) Analysis of GCase activity in HEK-293 cells exposed to ROT (0, 10, and 50 μM). (**D**) Representative CB-Dock2 3D images showing the molecular docking of GCase (PDB: 6T13) with ROT (PubChem CID 6758). The data are expressed as mean ± SD; * *p* < 0.05; ** *p* < 0.01; ns—not significant. Bars represent 1 out of 3 independent experiments (n = 3).

**Figure 2 ijms-24-10589-f002:**
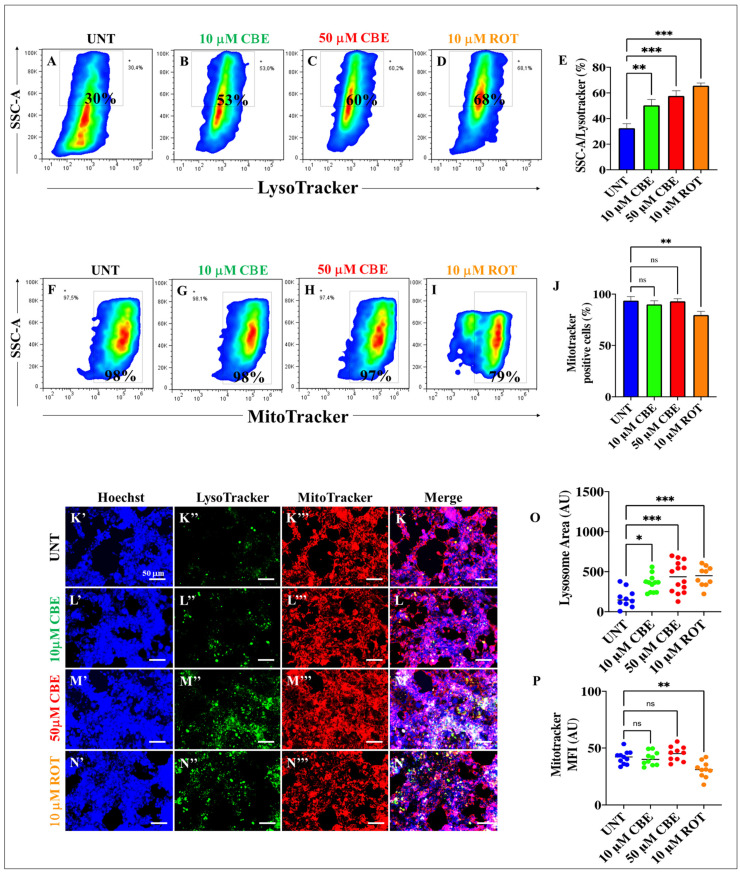
Rotenone (ROT) and conduritol-β-epoxide (CBE) induce accumulation of lysosomes, but rotenone diminishes mitochondrial membrane potential (ΔΨ_m_) only in HEK-293 cells. Representative density 2D plots showing SSC-A/LysoTracker analysis (lysosomes) in untreated cells (**A**) or treated with (10 μM) CBE (**B**), (50 μM) CBE (**C**), or (10 μM) ROT (**D**). Quantitative analysis of SSC-A/LysoTracker-positive cells (**E**). Representative density 2D plots showing SSC-A/MitoTracker analysis of untreated cells (**F**) or cells treated with (10 μM) CBE (**G**), (50 μM) CBE (**H**), or (10 μM) ROT (**I**). Quantitative analysis of SSC-A/MitoTracker-positive cells (**J**). The formation of acidic vacuoles was determined as described in Section 4. The percentage is the number of events for positive staining for acidic vacuoles in the upper-left quadrants (**A**–**D**,**F**–**I**), and color contrast indicates cell population density: dark blue < light blue < green < yellow < red. Representative fluorescence images showing Hoechst (**K′**–**N′**), LysoTracker (**K″**–**N″**), MitoTracker (**K‴**–**N‴**), and merge (**K**–**N**) of untreated HEK-293 cells (**K**) or cells treated with (10 μM) CBE (**L**), (50 μM) CBE (**M**), or (10 μM) ROT (**N**). Quantitative analysis of LysoTracker-stained area (**O**). Quantitative analysis of MitoTracker mean fluorescence intensity (**P**). The data are expressed as mean ± SD; * *p* < 0.05; ** *p* < 0.01; *** *p* < 0.001; ns—not significant. The smooth dot plots, bars, and photomicrographs represent 1 out of 3 independent experiments (n = 3). Image magnification, 20×.

**Figure 3 ijms-24-10589-f003:**
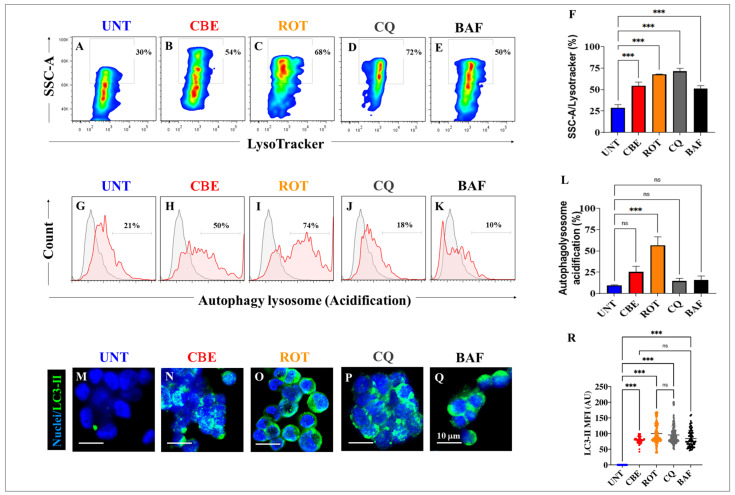
Rotenone (ROT) and conduritol-β-epoxide (CBE) induce accumulation of lysosomes and phagolysosomes in HEK-293 cells. Representative density 2D plots showing SSC-A/LysoTracker analysis (lysosomes) in untreated cells (**A**) or cells treated with (10 μM) CBE (**B**), (10 μM) ROT (**C**), (10 μM) chloroquine (CQ) (**D**), or (10 nM) bafilomycin A1 (BAF) (**E**). Quantitative analysis of SSC-A/LysoTracker-positive cells (**F**). Representative density 2D plots showing the autophagy–lysosome acidification of untreated cells (**G**) or cells treated with (10 μM) CBE (**H**), (10 μM) ROT (**I**), (10 μM) chloroquine (CQ) (**J**), or (10 nM) bafilomycin A1 (BAF) (**K**). Quantitative analysis of autophagy–lysosome-acidification-positive cells (**L**). The formation of acidic vacuoles was determined as described in Section 4. The percentage is the number of events for positive staining for acidic vacuoles in the upper-left quadrants (**A**–**E**), and color contrast indicates cell population density: dark blue < light blue < green < yellow < red. Representative immunofluorescence images showing LC3-II reactivity in untreated cells (**M**) or cells treated with (10 μM) CBE (**N**), (10 μM) ROT (**O**), (10 μM) chloroquine (CQ) (**P**), or (10 nM) bafilomycin A1 (BAF) (**Q**). Quantitative analysis of LC3-II mean fluorescence intensity (**R**). The data are expressed as mean ± SD; *** *p* < 0.001; ns—not significant. The dot plots, bars, histograms, and photomicrographs represent 1 out of 3 independent experiments (n = 3). Image magnification, 200×.

**Figure 4 ijms-24-10589-f004:**
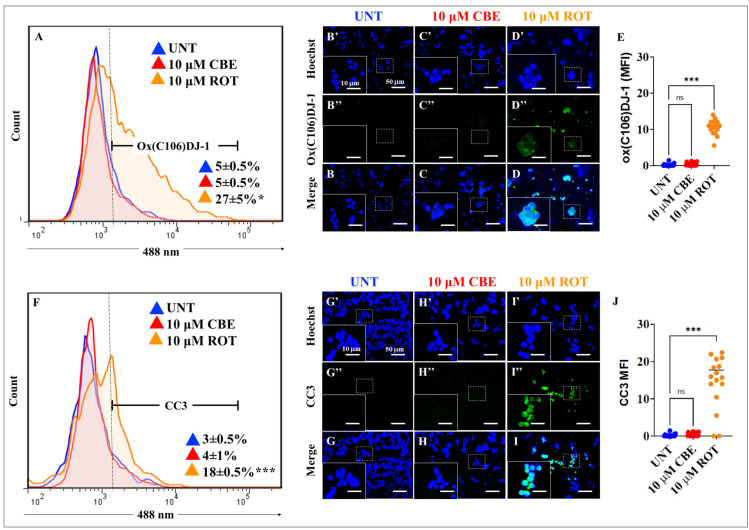
Rotenone (ROT) but not conduritol-β-epoxide (CBE) induces oxidation of DJ-1 proteins at residue Cys^106^ and cleaved caspase 3 (CC3). (**A**) Representative flow cytometry histogram analysis showing the oxDJ-1(Cys^106^)-positive population in untreated cells (blue curve) or cells treated with (10 μM) CBE (red) or (10 μM) ROT (orange). Representative fluorescence images showing Hoechst (**B′**–**D′**), oxDJ-1(Cys^106^)-positive (**B″**–**D″**), and merge (**B**–**D**) in untreated HEK-293 cells (**B**) or cells treated with (10 μM) of CBE (**C**) or (10 μM) ROT (**D**). Quantitative analysis of ox(Cys^106^) DJ-1 mean fluorescence intensity (**E**). Representative flow cytometry histogram analysis showing the CC3-positive population in untreated (blue), (10 μM) CBE (red)-, or (10 μM) ROT (orange)-treated cells (**F**). Representative fluorescence images showing Hoechst (**G′**–**I′**), CC3-positive (**G″**–**I″**), and merge (**G**–**I**) of untreated HEK-293 cells (**G**) or cells treated with (10 μM) CBE (H) or (10 μM) ROT (**I**). Quantitative analysis of CC3 mean fluorescence intensity (**J**). The data are expressed as mean ± SD; * *p* < 0.05; *** *p* < 0.001; ns—not significant. The histograms, bars, and photomicrographs represent 1 out of 3 independent experiments (n = 3). Image magnification, 20×. White line (area) represents magnification of broken line (area).

**Figure 5 ijms-24-10589-f005:**
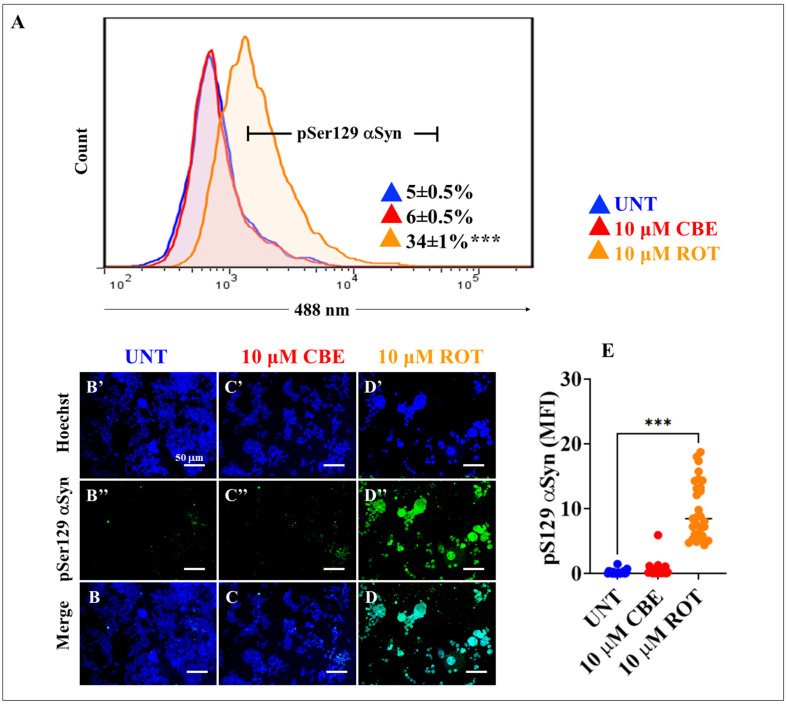
Rotenone (ROT) but not conduritol-β-epoxide (CBE) induces **α**-synuclein (α-Syn) phosphorylation at residue Ser^129^ and LRRK2 kinase in HEK-293 cells. (**A**) Representative flow cytometry histogram analysis showing the pSer^129^α-Syn-positive population in untreated cells (blue) or cells treated with (10 μM) CBE (red) or (10 μM) ROT (orange). Representative fluorescence images showing Hoechst (**B′**–**D′**), pSer^129^α-Syn-positive (**B″**–**D″**), and merge (**B**–**D**) in untreated HEK-293 cells (**B**) or cells treated with (10 μM) CBE (**C**) or (10 μM) ROT (**D**). Quantitative analysis of pSer^129^ α-Syn mean fluorescence intensity (**E**). Representative flow cytometry histogram analysis showing the pS^935^ LRRK2-positive population untreated (blue) or treated with (10 μΜ) CBE (red) or (10 μM) ROT (orange) (**F**). Representative fluorescence images showing Hoechst (**G′**–**I′**), pS^935^ LRRK2-positive (**G″**–**I″**), and merge (**G**–**I**) of untreated HEK-293 cells (**I**) or cells treated with (10 μM) CBE (**H**) or (10 μM) ROT (**I**). Quantitative analysis of pS^935^ LRRK2 mean fluorescence intensity (**J**). The data are expressed as mean ± SD; *** *p* < 0.001; ns—not significant. The histograms, dot graphs, and photomicrographs represent 1 out of 3 independent experiments (n = 3). Image magnification, 20×.

**Figure 6 ijms-24-10589-f006:**
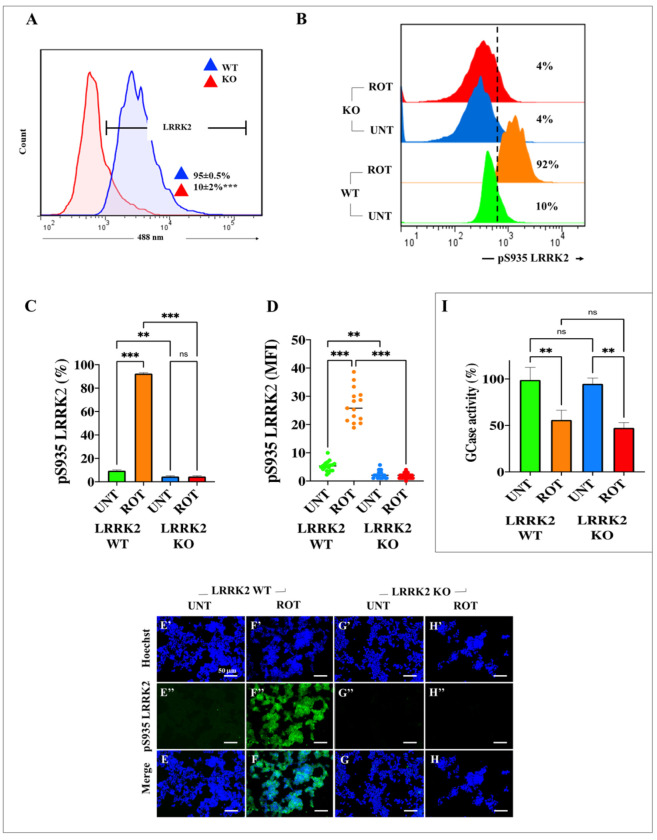
Rotenone (ROT) does not induce the phosphorylation of LRRK2 in HEK-293 LRRK2 KO cells but reduces the levels of glucocerebrosidase (GCase) activity in both HEK-293 LRRK2 WT and KO cells. Representative flow cytometry histogram analysis showing the total LRRK2-positive population of HEK-293 LRRK2 WT (blue) and KO cells (red) (**A**). Quantitative analysis of pS^935^ LRRK2-positive cells untreated (green, blue) or treated with (10 μM) ROT (orange, red) in HEK-293 LRRK2 WT and KO cells (**B**). Percentage of pS^935^ LRRK2-positive cells untreated (green, blue) or treated with (10 μM) ROT (orange, red) in HEK-293 LRRK2 WT and KO cells (**C**). Quantitative analysis of pS^935^ LRRK2 mean fluorescence intensity (**D**). Representative fluorescence images showing Hoechst (**E′**–**H′**), pS^935^ LRRK2-positive (**E″**–**H″**), and merge (**E**–**H**) in HEK-293 LRRK2 WT cells (**E**,**F**) and KO cells (**G**,**H**) untreated (**E,G**) or treated with (10 μM) of ROT (**F**,**H**). Analysis of GCase activity in HEK-293 LRRK2 WT and KO cells without or with (10 μM) ROT (**I**). The data are expressed as mean ± SD; ** *p* < 0.01; *** *p* < 0.001. The histograms, bars, and photomicrographs represent 1 out of 3 independent experiments (n = 3). Image magnification, 20×.

**Figure 7 ijms-24-10589-f007:**
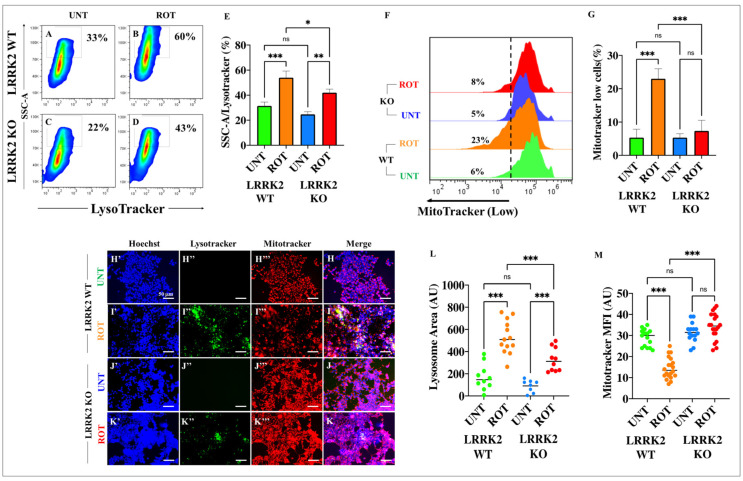
Rotenone (ROT) induces the accumulation of lysosomes but does not reduce mitochondrial membrane potential (ΔΨ_m_) in HEK-293 LRRK2 KO cells. Representative density 2D plots showing SSC-A/LysoTracker analysis (complexity of lysosomes) of untreated HEK-293 LRRK2 WT cells (**A**) or cells treated with (10 μM) ROT (**B**), and HEK-293 LRRK2 KO cells untreated (**C**) or treated with (10 μM) ROT (**D**). Quantitative analysis of SSC-A/LysoTracker-positive cells (**E**). The formation of acidic vacuoles was determined as described in Section 4. The percentage is the number of events for positive staining for acidic vacuoles in the upper-left quadrants (**A**–**D**), and color contrast indicates cell population density: dark blue < light blue < green < yellow < red. Representative flow cytometry histograms showing MitoTracker analysis of untreated or treated with (10 μM) ROT HEK-293 LRRK2 WT and KO cells. (**F**). Quantitative analysis of MitoTracker-depleted cells (**G**). Representative fluorescence images showing Hoechst (**H′**–**K′**), LysoTracker (**H″**–**K″**), MitoTracker (**H‴**–**K‴**), and merge (**H**–**K**) HEK-293 LRRK2 WT and KO cells untreated (**H**,**J**) or treated (**I**,**K**) with (10 μM) ROT. Quantitative analysis of LysoTracker-stained area (**L**). Quantitative analysis of MitoTracker mean fluorescence intensity (**M**). The data are expressed as mean ± SD; * *p* < 0.05; ** *p* < 0.01; *** *p* < 0.001; ns—not significant. The smooth dot plots, bars, histograms, and photomicrographs represent 1 out of 3 independent experiments (n = 3). Image magnification, 20×.

**Figure 8 ijms-24-10589-f008:**
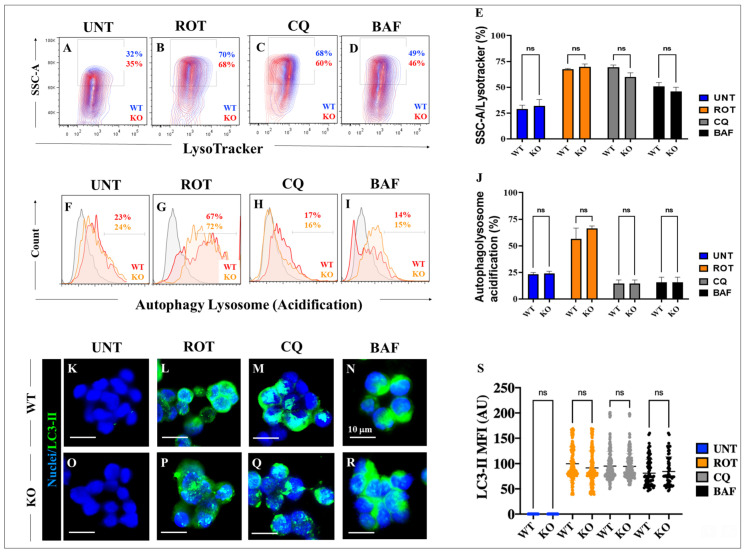
Rotenone (ROT) induces accumulation of the autophagolysosome flux in HEK-293 LRRK2 KO cells. Representative density 2D plots showing SSC-A/LysoTracker analysis (granular content or complexity of lysosomes) of untreated HEK-293 LRRK2 WT or HEK-293 LRRK2 KO cells (**A**) or cells treated with 10 μM ROT (**B**), (10 μM) chloroquine (CQ) (**C**), or (10 nM) bafilomycin A1 (BAF) (**D**). Quantitative analysis of SSC-A/LysoTracker-positive cells (**E**);. The formation of acidic vacuoles was determined as described in Section 4. The percentage is the number of events for positive staining for acidic vacuoles in the upper-left quadrants (**A**–**D**), and color indicates cell population density of HEK-293 LRRK2 WT (blue) and HEK-293 LRRK2 KO (red) cells. Representative flow cytometry histograms showing the autophagy–lysosome acidification of untreated HEK-293 LRRK2 WT or KO cells (**F**) or cells treated with 10 μM ROT (**G**), (10 μM) chloroquine (CQ) (**H**), or (10 nM) bafilomycin A1 (BAF) (**I**). Quantitative analysis of autophagy–lysosome-acidification-positive cells (**J**). The percentage is the number of events for positive staining for acidic vacuoles, and color indicates cell population of HEK-293 LRRK2 WT (red) and HEK-293 LRRK2 KO (orange) cells. Representative immunofluorescence images showing LC3-II accumulation in HEK-293 LRRK2 WT (**K**–**N**) and KO cells (**O**–**R**) untreated (**K**,**O**) or treated with (10 μM) ROT (**L**,**P**), (10 μM) chloroquine (CQ) (**M**,**Q**), or (10 nM) bafilomycin A1 (BAF) (**N**,**R**). Quantitative analysis of the accumulation of LC3-II as mean fluorescence intensity (**S**). The data are expressed as mean ± SD; ns—not significant. The contour diagrams, histograms, bars, dot graphs, and photomicrographs represent 1 out of 3 independent experiments (n = 3). Image magnification, 200×.

**Figure 9 ijms-24-10589-f009:**
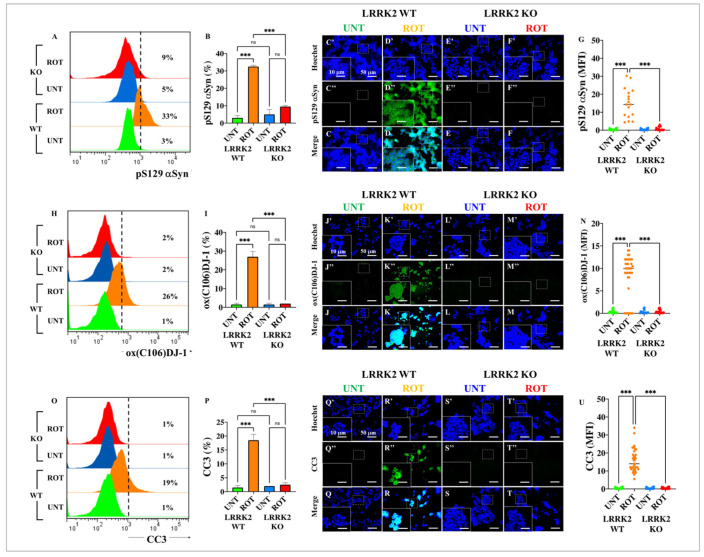
Rotenone (ROT) neither induces phosphorylation of α-synuclein (α-Syn), oxidation of DJ-1 protein at residue Cys^106^ nor generates cleaved caspase 3 (CC3). Representative flow cytometry histogram analysis showing the α-synuclein (α-Syn)-positive population in HEK-293 LRRK2 WT (blue) or KO cells (red) (**A**). Quantitative analysis of α-Syn (**B**). Representative fluorescence images showing Hoechst (**C′**–**F′**), α-Syn (**C″**–**F″**), and merge (**C**–**F**) HEK-293 LRRK2 WT and KO cells untreated (**C**,**E**) or treated (**D**,**F**) with (10 μM) of ROT. Quantitative analysis of α-Syn-stained area (**G**). Representative flow cytometry histogram analysis showing the oxDJ-1Cys^106^-positive population in HEK-293 LRRK2 WT (blue) or KO cells (red) (**H**). Quantitative analysis of oxDJ-1Cys^106^ (**I**). Representative fluorescence images showing Hoechst (**J′**–**M′**), oxDJ-1Cys^106^ (**J″**–**M″**), and merge (**J**–**M**) HEK-293 LRRK2 WT and KO cells untreated (**J**,**L**) or treated (**K**,**M**) with (10 μM) ROT. Quantitative analysis of oxDJ-1Cys^106^-stained area (**N**). Representative flow cytometry histogram analysis showing the cleaved caspase 3 (CC3)-positive cell population in HEK-293 LRRK2 WT (blue) or KO cells (red) (**O**). Quantitative analysis of CC3 (**P**). Representative fluorescence images showing Hoechst (**Q′**–**T′**), oxDJ-1Cys^106^ (**Q″**–**T″**), and merge (**Q**–**T**) HEK-293 LRRK2 WT and KO cells untreated (**Q**,**S**) or treated (**R**,**T**) with (10 μM) ROT. Quantitative analysis of CC3-stained area (**U**). The data are expressed as mean ± SD; *** *p* < 0.001; ns—not significant. The smooth dot plots, bars, histograms, and photomicrographs represent 1 out of 3 independent experiments (n = 3). Image magnification, 20×. White line (area) represents magnification of broken line (area).

**Figure 10 ijms-24-10589-f010:**
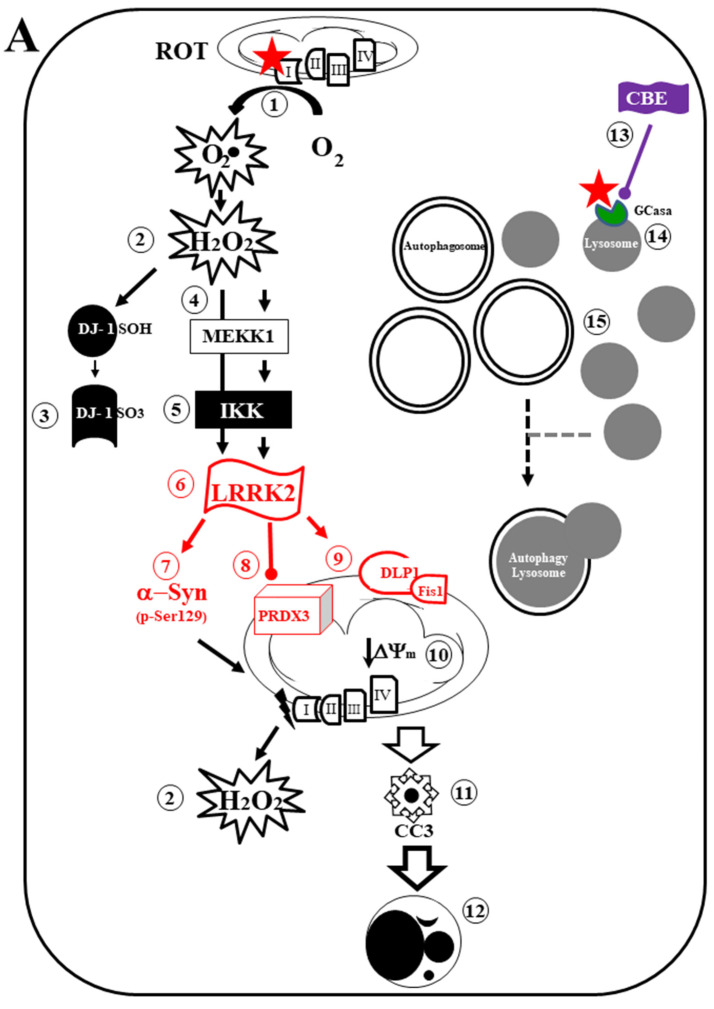
Schematic model of cell signaling induced by rotenone and conduritol-β-epoxide: A mechanistic explanation of the interaction between LRRK2 kinase, α-Synuclein, glucocerebrosidase, lysosomes, and autophagosomes. (**A**) Rotenone (ROT, red full star) binds to the ubiquinone binding site of mitochondrial complex I (NADH:ubiquinone oxidoreductase), thus preventing electron transfer via Flavin mononucleotide (FMN) to coenzyme Q10 (**1**). Consequently, the interruption of the electron transport chain concomitantly generates anion superoxide (O_2_^−^) and hydrogen peroxide (H_2_O_2_, **2**). This last compound is capable of oxidizing the stress sensor protein DJ-1Cys^106^-SH into DJ-1Cys^106^-SO_3_ (**3**), directly activates LRRK2 (leucine-rich repeat kinase 2) kinase by autophosphorylation (**4**) or indirectly phosphorylates LRRK2 through activation of MEKK1 (mitogen-activated protein/extracellular signal-related protein kinase (MAP/ERK) kinase (MEK))/IKK (IκB kinase, **5**). Once LRRK2 is phosphorylated at Ser^935^, the active p-(S-^935^)-LRRK2 kinase (**6**) phosphorylates three major targets: (i) alpha-synuclein (α-Syn) at residue Ser^129^ (**7**), which, in turn, interacts with mitochondria complex I, thereby generating H_2_O_2_, and induces loss of mitochondrial membrane potential (ΔΨ_m_); (ii) it inactivates protein PRDX3 (peroxiredoxin 3, **8**), thereby preventing H_2_O_2_ catalysis; (iii) p-(S-^935^)-LRRK2 activates the mitochondrial fission protein DLP-1 (dynamin-like protein 1, **9**), which, together with the fission protein-1 (Fis-1) receptor, induces mitochondria depolarization, fragmentation, and aggregation (**10**). Subsequently, the release of apoptogenic proteins (e.g., cytochrome C) results in the production of cleaved caspase 3 (**11**), which is responsible for chromatin condensation and DNA fragmentation (**12**) in HEK-293 LRRK2 WT cells. The nucleus morphology constitutes the typical hallmark of apoptosis. Alternatively, ROT and conduritol-β-epoxide (CBE, **13**) bind to the enzyme glucocerebrosidase (GCase) (**14**). The reduced catalytic activity of GCase results in a limited fusion of autophagosomes and lysosomes, leading to their respective accumulation (**15**). (**B**) Rotenone (ROT, red full star) binds to the complex I (**1**), thereby generating (O_2_^.−^) and hydrogen peroxide (H_2_O_2_, **2**). This last compound is decomposed by PRDX3 (**8**). As a result, ΔΨ_m_ is preserved (**16**), avoiding the release of apoptogenic proteins. Therefore, the nucleus is conserved intact (**17**) in HEK-293 LRRK2 KO cells. Additionally, ROT binds to GCase (**14**), resulting in the accumulation of lysosomes and autophagy–lysosomes (**15**). The cell shows neither signs of oxidative stress (OS) nor apoptosis markers.

**Table 1 ijms-24-10589-t001:** In silico molecular docking analysis of conduritol-β-epoxide (CBE), rotenone (ROT), and glucocerebrosidase (GCase).

Submitted Protein *	Submitted Ligand **	Vina Score ***	Cavity Volume (Å^3^)	Center (x, y, z)	Docking Size (x, y, z)	Contact Residue
**GCase** **(6T13)**	CBE (conformer 3D CID 9989541)	**−6.0**	3950	13, 10, −2	16, 34, 33	Chain A: **Asp^127^** **Phe^128^** **Trp^179^** **Asn^234^** **Glu^235^** **Tyr^244^** Phe^246^ **Gln^284^ Tyr^313^ Glu^340^ Cys^342^ Ser^345^ Trp^381^ Asn^396^** Val^398^
	ROT (conformer 3D CID 6758)	**−9.2**	3950	13, 10, −2	22, 34, 33	Chain A: **Asp^127^ Phe^128^ Trp^179^ Asn^234^ Glu^235^** Ser^237^ Ala^238^ Leu^241^ **Tyr^244^** Pro^245^ **Phe^246^** Asp^283^ **Gln^284^ Tyr^313^** Leu^314^ **Glu^340^ Cys^342^ Ser^345^ Trp^381^ Asn^396^**

* According to RCSB Protein Data Bank (https://www.rcsb.org/; accessed in June 2023). ** According to PubChem database (https://pubchem.ncbi.nlm.nih.gov/; accessed in June 2023). *** According to CB-dock2: An accurate protein–ligand blind docking tool (https://cadd.labshare.cn/cb-dock2/php/index.php; accessed in June 2023). Bold letters represent similar amino acid residues in the catalytic pocket of the protein that interacts with CBE and ROT.

## Data Availability

All relevant data are within the manuscript.

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
