# Peer review of "Rotenone Blocks the Glucocerebrosidase Enzyme and Induces the Accumulation of Lysosomes and Autophagolysosomes Independently of LRRK2 Kinase in HEK-293 Cells"

_ijms, 2023, doi:10.3390/ijms241310589_

Round 1

Reviewer 1 Report

See above

Reviewer 2 Report

Summary

The manuscript submitted by Dr. Marlene Jimenez-Del-Rio and co-workers report changes in glucocerebrosidase activity and accumulation of lysosomes and autophagolysosomes during the administration of Rotenone and conduritol-epoxide in HEK-293 cell cultures. Interestingly, these parameters do not depend on the presence of active LRRK2 kinase. However, other parameters, including the mitochondrial membrane potential, do depend on active LRRK2 genes. Based on these results, the authors discuss the effects of mitochondrial damage, p-Ser129α-Syn, reduced GCase activity, and LRRK2 phosphorylation on the autophagy lysosomal pathway and apoptosis. Furthermore, they suggest that LRRK2 and GCase may be useful therapeutic targets.

While the experiments are conducted carefully and reported with appropriate statistical treatment of the results, the manuscript needs work (see below).

Major points

The manuscript needs a major rework:

·      A summary chart of the results with an indication of their relation to published work is needed. Such a chart would simplify the text, especially in the discussion, making the report accessible to non-afficionados.

·      Reduce repetitions of technical matters that have already been addressed in the Results and Methods sections.

·      The Conclusions should contain text to return to the conceptual contributions to the understanding of Parkinson's Disease highlighted in the Abstract

Minor points

·      Several figures need a more detailed explanation. For example,

o   What are the boxes and numbers in several flow cytometry results (e.g., Figure 2AD and F-I).

o   An explanation of the colors in the cell maps of SSC-A in several figures is missing.

·      Add an explanation/reference in the Methods of determining membrane potential.

·      Update references, e.g., to the importance of Glu340 in CBE (line 156)

·      Line 65: a single component cannot “interact” (dictionary explanation of “interact”: act in such a way as to have an effect on each other)

·      Line 79: define NLC.

·      Line 194: and>through

·      Line 194: unperturbed>did not perturb.

See above

Reviewer 3 Report

Perez-Abshana et al., in their manuscript describe the complex relationship between rotenone (ROT), the lysosomal enzyme glucocerebrosidase (GCase), and the kinase leucine-rich repeat serine/threonine protein kinase 2 (LRRK2) and their involvement in Parkinson’s disease. Parkinson’s disease is a neurodegenerative disorder that affects patients motor abilities. With the increasing number of people suffering from this disease, there is a need to better understand the molecular basis of the disease to find a better therapeutic target.

The work done by the authors is immense and is done with great attention to detail. It is, however, confusing to read the manuscript in some areas.

The following are my comments.

1] Figure 3: In this figure, the authors are trying to study if ROT, that is used to model PD, causes accumulation of autophagic lysosomes. While treatment with CBE and ROT alone shows the accumulation of autophagic lysosomes, a co-treatment of HEK293 cells+ ROT or CBE+ CQ, HEK293 cells+ ROT or CBE+ BAF will be a better comparison to show the effect of ROT or CBE on lysosome and autophagic lysosome accumulation. This might help understand a direct link, if there is any, between the role played by my ROT and CBE in lysosome or autophagy-lysosome function.

2] Fig5: The effect of ROT on LRRK2 phosphorylation would be better if quantified by western blot analysis.

3] Fig 8: As stated above, the co-treatment of LRRK2 KO and WT cells with ROT+ CQ and ROT+ BAF might help understand the link between ROT and LRRK2 and autophagic lysosomal mediated processing.

4] It will greatly help readers if there was a schematic representation of the proposed mechanism of interaction between ROT- LRRK2- GCase- and the process of autophagy and the accumulation of the alpha-synuclein in PD pathogenesis.

Many results presented in this paper confirm some earlier findings that have been published.  But this paper makes one addition to the current knowledge about PD disease, which is that LRRK2, the main kinase implicated in PD pathogenesis, may not directly affect the autophagy-lysosome function that is known to cause the accumulation of alpha-synuclein in the disease.

The quality of the language in this paper is good.

Reviewer 4 Report

Comments to Authors

The manuscript entitled, “Rotenone Blocks the Glucocerebrosidase Enzyme and Induces the Accumulation of Lysosomes and Autophagolysosomes Independently of LRRK2 Kinase in HEK-293 Cells” by Perez-Abshana et al., showed that ROT decreases the activity of GCase, and results in the accumulation of lysosomes and autophagolysosomes in HEK-293 cells by conducting experiments in HEK-293 cells and LRKK2 knockout HEK-293 cells, along with a GCase inhibitor (CBE). The major weakness is that the study was conducted in kidney-derived cells instead of neuronal-derived cells, as substrates (e.g., glucocerebroside versus galactocerebroside) and enzyme isoforms vary between cell types. The paper should address the following points:

Provide a rational for using HEK-293 versus a neuronal-derived cell line.

The levels of galactocerebroside are much higher than glucocerebroside in neuronal cells. If this is the case, would less activity of GCase still result in accumulation of lysosome and autophagolysosomes in neuronal-derived cell lines. This point should be discussed to strengthen the correlation of the study with PD. 

The results need improvement since the text of the results does not describe the data acquired in each panel. For instance, text of the IMF data is lacking and more stated as if they are supplementary figure. This needs to be addressed.

For sake of clarity and reduction in panels per figure, just show 1 representative Western blot and put the others in supplementary section.

If you are defining abbreviations then use them.

Sub-headings need to be clarified such 2.2, 2.3 and figure legend caption for figure 6.

Fig 2B result matches fig3 B result, yet concentration of CBE is different. Please verify concentration of CBE used.

Lines 234-236, do you have stats to claim this significance.

Clarify why autophagy lysosome acidification was increased with ROT treatment but not CBE (Fig 3G-L).

Clarify lines 328-332. Also panel Fig6K is ignored.

It would be helpful to rephrase lines 366-368.

Figure 7E ROT increases both wt and ko and text of results should state this observation.

Round 2

Reviewer 1 Report

The modifications implemented in the manuscript have improved its overall quality. Although the authors express concerns that renaming the individual panels of the figures may compromise result clarity, I posit that renaming the panels (as suggested by the authors in the table) and eliminating the black squares/borders of the fgiures could enhance result interpretation, particularly when comparing different treatments. 

Reviewer 2 Report

Thank you for carefully addressing my concerns

Reviewer 3 Report

The authors have addressed my concerns and the inclusion of the the schematic figure further helps in the understanding of the manuscript. 
